# Population Structure of a Worldwide Collection of Tropical *Japonica* Rice Indicates Limited Geographic Differentiation and Shows Promising Genetic Variability Associated with New Plant Type

**DOI:** 10.3390/genes13030484

**Published:** 2022-03-09

**Authors:** Vikram Jeet Singh, Prolay Kumar Bhowmick, Kunnummal Kurungara Vinod, Subbaiyan Gopala Krishnan, Shekharappa Nandakumar, Amit Kumar, Manoj Kumar, Sonu Shekhawat, Brijesh Kumar Dixit, Ankit Malik, Ranjith Kumar Ellur, Haritha Bollinedi, Mariappan Nagarajan, Ashok Kumar Singh

**Affiliations:** 1Division of Genetics, ICAR-Indian Agricultural Research Institute, Pusa, New Delhi 110012, India; jeet2012vikram@gmail.com (V.J.S.); kkvinod@iari.res.in (K.K.V.); gopal.s@icar.gov.in (S.G.K.); nandus237@gmail.com (S.N.); ch.manojmalik@gmail.com (M.K.); sonushekhwat101@gmail.com (S.S.); brijeshdixit152@gmail.com (B.K.D.); ankitchunsa@gmail.com (A.M.); ranjith.ellur@icar.gov.in (R.K.E.); haritha.bollinedi@icar.gov.in (H.B.); director@iari.res.in (A.K.S.); 2ICAR Research Complex for NEH Region, Umiam, Meghalaya 793103, India; amit.kumar3@icar.gov.in; 3Rice Breeding and Genetics Research Center, ICAR-IARI, Aduthurai 612101, India; m.nagarajan@icar.gov.in

**Keywords:** population structure, new plant type, interspecific hybridization, genetic variability, rice, genetic diversity

## Abstract

Abating the approaching yield plateau in rice requires taking advantage of potential technologies that requires knowledge on genetic diversity. Hybrid breeding, particularly in *indica* rice, requires the recruitment of large genetic variability from outside because the available genetic diversity of the cultivated pool has already been utilized to a great extent. In this study, we examined an assembly of 200 tropical *japonica* lines collected worldwide for population genetic structure and variability in yield-associated traits. Tested along with 30 *indica* and six wild rice lines belonging to India, the tropical *japonica* lines indicated great phenotypic variability, particularly related to new plant type (NPT) phenology, and formed six clusters. Furthermore, a marker-based characterization using a universal diversity marker panel classified the genotype assembly into four clusters, of which three encompassed tropical *japonica* lines, while the last cluster included mostly *indica* lines. The population structure of the panel also revealed a similar pattern, with tropical *japonica* lines forming three subpopulations. Remarkable variation in the allelic distribution was observed between the subpopulations. Superimposing the geographical sources of the genotypes over the population structure did not reveal any pattern. The genotypes sourced closer to the center of origin of rice showed relatively little diversity compared with the ones obtained from other parts of the world, suggesting migration from a common region of origin. The tropical *japonica* lines can be a great source of parental diversification for hybrid development after confirming the presence of widely compatible genes.

## 1. Introduction

In India, as the major food crop for more than 1.25 billion people, rice is grown in a 43.40-million-hectare (mha) area, with a paddy production of 157.20 million tones (mt) [1]. Twenty percent of the world’s rice production is contributed by India [2], which forms a major segment of Asia’s global share, amounting to about 670 million tonnes [3,4]. Although first in rice acreage, India ranks second after China in production. China produces 211.1 mt from 30.44 mha with a productivity of 69.32 q/ha compared with India’s productivity of 23.9 q/ha [5,6].

Among the two subspecies of the cultivated rice (*Oryza sativa*), *indica* and *japonica* have become genetically isolated in the evolutionary process due to their strong cross hybridization barrier. This has resulted in the containment of genetic diversity within each subspecies. Although predominant in the rice gene pool *indica* cultivars are relatively less productive than the *japonica*, *japonica* genotypes are relatively more diverse than *indica* genotypes. The Indian rice, which is majorly composed of *indica*, types is adapted to the tropical environment, while the *japonica* rice includes two subtypes: temperate and tropical (upland). Currently common in China and other southeast Asian nations, *japonica* rice prefers a cooler climate. The tropical *japonica* (TRJ), also known as *bulu* rice, is an intermediate type between *indica* and *japonica*. Considered previously as the third subspecies *javanica*, this group of genotypes can hybridize with both *indica* and *japonica* by overcoming the crossability barrier. Therefore, tropical *japonica*s are considered the bridge for *indica*–*japonica* hybridization and a source for increasing genetic diversity in rice. Tropical *japonica*, have fewer tillers, sturdy culm, and vigorous root architecture.

Being genetically less diverse, rice has a low average level of heterosis [7]. Therefore, merging the genetic diversity across subspecies could bring in increased genetic diversity and therefore increased heterosis, for which tropical *japonica* would be useful. Therefore, the genetic diversity in the tropical *japonica* rice germplasm can be of great advantage for two reasons: (i) as one of the parents in the hybrid development of tropical *japonica*s hybridizing both *indica* and *japonica* lines and (ii) as the bridge between true *indica* and *japonica* parents augmenting additional diversity during parental population improvement. Nevertheless, the use of tropical *japonica* in hybrid rice breeding has been limited due to their lower productivity and other undesirable traits [8]. Furthermore, the information on their natural diversity among these group of genotypes is restricted and is less characterized at the molecular level [9]. An assessment of genetic diversity can be carried out using morphological, biochemical, and molecular markers. This would aid in selecting diverse parental lines for hybrid development. Morphological selection is reasonable for the exploitation of quantitative traits and often marred by the high influence of the environment. In this context, molecular-marker-based diversity is more reliable as it is free from environmental influence. Thus, markers help in the precise characterization of genome diversity, identifying novel alleles and facilitating the development of new breeding lines with desirable traits. A wide range of marker systems are now available, among which SSRs are commonly used due to their abundance, genome-wide distribution, and high polymorphism [10,11,12].

Genotypes accumulate several viable mutations during the evolutionary process, which forms the basis of genetic diversity. Furthermore, forces such as recombination, random drift, natural selection, etc., shape the genetic structure of populations. In the recent past, understanding the population structure has become a feature of great interest, as this can help in thte selection of diverse parents and in mapping marker–trait associations. As a tool, an analysis of population structure can estimate the similarity levels among the individuals, subpopulations, as well as admixtures. When the samples are drawn with diverse geographical origins, an analysis of population structure indicates the pattern of geographical distribution among the populations. In practice, the most common approaches to stratifying population structure are model-based predictions and principal component analysis (PCA). The model-based predictions help in estimating the co-ancestry coefficients for defining population structure, whereas the PCA helps in reducing the multi-dimensional data into principal components that are orthogonal and independent, thereby providing opportunities to visualize the distribution pattern of the individuals [13].

In this study, we characterized a set of tropical *japonica* lines collected worldwide, using SSR markers and agro-morphological traits along with 23 *indica,* 4 *aus*, 3 Basmati, and 6 wild rice genotypes and estimating the phenotypic and genotypic diversity and population structure. The quantification of genetic variability for agronomic traits has a particular focus on yield and its components. We aim to use this information towards identifying potential parental sources in future hybrid rice development, particularly *indica*–*japonica*-based high yielding hybrids.

## 2. Materials and Methods

### 2.1. Plant Materials and Experimentation

A total of 200 tropical *japonica* genotypes originating from 43 countries (Appendix A) were maintained as International Rice Germplasm Collection (IRGC) at International Rice Research Institute (IRRI), Los Baños, Laguna, Philippines. The materials were maintained at Division of Genetics, ICAR-Indian Agricultural Research Institute (ICAR-IARI), New Delhi. Details of the lines used are provided in Appendix A. Plant materials were field evaluated for three successive seasons at ICAR-IARI, New Delhi, in a randomized complete block design with two replications. Each entry was grown in a plot of size 1.2 m^2^, keeping plant-to-plant and row-to-row distances of 15 cm and 20 cm, respectively. The entire experiment was managed with recommended agronomic practices.

### 2.2. Phenotyping

At the time of physiological maturity, observations of the morphological traits (tiller number, plant height, and panicle length) were collected from five random plants. These plants were harvested, shade-dried, and threshed to collect data on the single plant yield, number of spikelets per panicle, filled grains per panicle, and spikelet fertility (%). The agro-morphological traits were subjected to an analysis of variance (ANOVA). A principal component analysis (PCA) was performed to elucidate divergence among the tropical *japonica* genotypes. Eigenvalues of more than one was considered for identifying significant principal components. From the eigenvectors, significant traits that had a high contribution to major principal components were identified. PCA scores were used for hierarchical clustering using squared Euclidean distances. The biplot was generated using the first pair of major components to explore similarities and patterns among the tropical *japonica* genotypes. All of the analyses were carried out under R statistical environment [14]. 

### 2.3. Genotyping

Fifty microsatellite markers (Appendix A) from the GCP panel [15] of microsatellite markers, which are evenly distributed across 12 chromosomes, were used for the estimation of genetic relation among 200 tropical *japonica* lines along with thirty-six different checks (*indica*, *aus*, Basmati, short grain aromatic, wild species, and northeastern cultivars). Three-week-old seedlings were used to isolate genomic DNA using a standard protocol [16]. A reaction volume of 10 µL was prepared to perform the polymerase chain reaction (PCR). Initially, at a temperature of 94 °C, the template DNA was denatured (5 min); then, the 35 cycles of PCR amplification consisted of three steps, denaturation at 94 °C for 30 s, annealing for 1 min at 55 °C and primer extension for 1 min at 72 °C, and a final extension for 7 min at 72 °C; and then, the product was cooled to 4 °C. The amplicons were separated using 3.5% agarose gel prepared using TAE buffer (1×). A 50 base pair DNA ladder was used to score the amplified fragments.

### 2.4. Diversity Analysis

The statistical analysis consisted of estimating alleles/locus, major allele frequency, expected and observed heterozygosity, gene diversity, and polymorphic information content (PIC). Polymorphic information content was calculated as per the formula given by [17] for self-pollinated crops and further modified by [18].
$${\mathrm{PIC}}_{i}=\overset{n}{\underset{j=1}{\sum }}P_{ij}^{2}$$
where P*_ij_* = frequency, *j* = allele, and *i* = marker.

The genetic distance between the tropical *japonica* accessions was computed using Rogers distance [19], and the distance matrix was subjected to hierarchical clustering using the unweighted pair group method using averages (UPGMA). The dendrogram was constructed using PowerMarker v.3.25 and visualized using MEGAX.

### 2.5. Determination of Population Structure

It is established that different gene frequencies among the individuals leads to stratification of the population into several subgroups. These differences may arise due to geographical isolation or any sort of selection performed on a population, i.e., artificial or natural. To determine the geographical effects on the population structure, pairwise F_ST_ values were computed between each geographical group. An analysis was performed using PopGene v.1.32 [20]. The pairwise F_ST_ values were used for Euclidean clustering using the hierarchical agglomerative method UPGMA and visualized using R statistical environment. Furthermore, to detect the cumulative population structure of the test panel, Bayesian modelling of the genotype data was used. The analysis assumes an admixture model for the population with correlated allelic frequencies. An analysis was carried out using Structure v.2.3.4. To detect the number of subpopulations, the structure generates a matrix of co-ancestry coefficients (Q matrix), which explains the probability of an individual falling into a subpopulation. The optimal number of subpopulations was determined using simulation summary statistics by determining the ΔK using an ad hoc procedure [21]. An analysis was performed using the online tool Structure Harvester [22]. Based on the estimated population structure, an analysis of molecular variance (AMOVA) was performed using Arlequin v.3.1 software [23]. Fixation indices were calculated as per Weir and Cockerham [24].

## 3. Results

### 3.1. Agro-Morphological Variability

Based on the analysis of variance (ANOVA) of the agronomic data, significant differences were noticed among the tropical *japonica* lines for traits such as plant height, tiller number, panicle length, total grains, spikelet fertility, and yield per plant (Table 1). 

Significant variation was observed for the genotypic and genotype × year interaction components. The year component, however, was found non-significant for all of the traits. The mean plant height among the genotypes was 127.3 cm, which ranged from 87.7 to 155.7 cm; tiller number ranged from 4.2 to 14.8, with a mean value of 8.0; and the panicle length ranged from 19.5 cm to 31.5 cm, with a mean value of 25.6 cm. Filled and unfilled grains per panicle ranged from 59.1 to 274.1 and from 5.6 to 118.7, with averages of 144.0 and 23.4, respectively. The total grains per panicle ranged from 67.2 to 323.8, and spikelet fertility ranged from 58.9 to 95.7%. The average grain yield per plant was 15.3 g, with a range from 5.0 to 34.1 g per plant. The agronomic traits showed normal distribution with near equal dispersion from the median (Figure 1 and Appendix A). Two traits, the number of unfilled spikelets per panicle and spikelet fertility, showed skewed distribution towards fertility. Traits such as tiller number, number of filled grains per panicle, total spikelets per panicle, and yield showed few extreme genotypes at the upper tail. However, the extreme values shown in unfilled spikelets contributed to lower tail extremes for spikelet fertility. Since the tropical *japonica* lines showed significant variation for phenotype expression, the diversity was estimated using the multivariate approach using PCA. 

### 3.2. Phenotypic Diversity of Tropical Japonica Lines

Among the eight measured traits, six were selected for phenotypic diversity analysis. Two traits, unfilled spikelets per panicle and filled grains per panicle, were removed from the analysis for being the constituent of the total grain number. Among the six principal components (PC) extracted by the PCA, three were identified as major as they possessed eigenvalues above 1.0. These components explained 75.8% of the total variation among the agronomic data. PC1 accounted for 36.0% variability followed by PC2 (21.2%) and PC3 (18.7%). Four traits showed a larger contribution towards PC1, followed by two traits for PC2 and one for PC3. The contribution of panicle length (31.8%) was the highest for PC1, followed by total grain number (27.9%), plant height (21.2%), and tiller number (13.8%) (Table 2). Yield (71.7%) and tiller number (12.1%) contributed majorly to PC2, while for PC3, the contribution of spikelet fertility (63.7%) was the highest, followed by tiller number (35.4%). A hierarchical clustering of the genotypic PCA scores using squared Euclidean distance revealed six distinct clusters. The maximum number of genotypes (173) was grouped into cluster I, which accounted for 86.5% of the total panel. The next group, cluster II, possessed 15 genotypes (7.5% of the population) and the third cluster had five genotypes (2.5% of the population). Cluster IV had the lowest number of genotypes (1) and accounted for just 0.5% of the panel, whereas cluster V and cluster VI carried 1.5% of the test panel each, comprising 15 genotypes per cluster (Figure 2). Most of the genotypes were tall and present in all of the clusters except for cluster IV (Table 3). The average tiller number was relatively high among the genotypes grouped into clusters V and VI. However, these clusters had only a few genotypes when compared with the first two clusters. Among these, three genotypes (IRGC 66529, IRGC 64656, and IRGC 64657) with more tillers than the rest were found grouped in cluster VI. Panicle length on average did not show significant variation between clusters. The number of total grains per panicle was the highest among clusters II and IV. Cluster II, in general, had a greater number of genotypes with a high total grain number, followed by the cluster 4 genotype IRGC56704. Nevertheless, three genotypes (IRGC 66757, IRGC 5554, and IRGC32704) possessing high total grain number were found in cluster I. The average spikelet fertility among the clusters was high for cluster I, followed by cluster V. However, cluster V had the highest average grain yield followed by cluster IV and cluster II. Genotypes such as IRGC328, IRGC11169, IRGC14530, and IRGC24275B with higher yield belonged to cluster I. The distribution of genotypes among the clusters did not show any specific country-wise pattern.

Among the 50 SSR markers used, 46 markers produced clear polymorphism among the test genotypes. Three markers were monomorphic, while one showed partial amplification. A total of 184 alleles were produced across 200 tropical *japonica* lines, with an average of four alleles per marker (Table 4). There were 3.83 markers per chromosome, producing 15.3 alleles on average. The number of alleles per marker ranged from two (OSR13, RM55, RM455, RM284, RM433, and RM484) to 11 (RM474). Among the 46 markers, six were bi-allelic (12%), sixteen were tri-allelic (32%), eleven were tetra-allelic (20%), eight were penta-allelic, two were hexa-allelic, and two were hepta-allelic (Appendix A). Among the chromosomes, a greater number of polymorphic markers were present on chromosome 1 than chromosome 8, with eight and seven markers, respectively. Chromosomes 3 and 5 had five markers each. Chromosomes 10 and 11 had four markers each, whereas chromosomes 6, 7, and 9 were represented by three markers each. The lowest polymorphic markers (two each) were found in chromosomes 2, 4, and 12. The average number of alleles per marker was highest (5.0) for chromosomes 10 and 11, followed by chromosome 5 (4.6) and chromosome 12 (4.5). The major allele frequency among the markers is the frequency of alleles with a proportion of more than 5%, which ranged between 0.58 in chromosome 2 and 0.86 in chromosome 4. Among the markers, major allele frequency ranged from 31% (RM334) to 99.2% (RM433). Correspondingly, the mean genetic distance was the lowest on chromosome 4 and was the highest on chromosome 2. Marker-based gene diversity ranged from 0.017 (RM433) to 0.74 (RM334), with a mean value of 0.42 (Appendix A). The observed heterozygosity (Ho), the number of individuals heterozygous per locus, ranged between 0.00 and 0.386 among the chromosomes. The mean polymorphism information content (PIC) was lowest on chromosome 4 (0.21) and highest in chromosomes 2 and 10. Among the markers, RM433 had the lowest PIC of 0.017, while RM334 had the highest PIC of 0.698, followed by RM474, with a PIC of 0.696.

### 3.3. Marker-Based Genetic Distance and Grouping

Genetic relatedness by Rogers’ genetic distance and the clustering using principal coordinate analysis revealed four major clusters (Figure 3). Cluster III had 44.9% of the genotypes, followed by clusters II, I, and IV, with 21.6%, 17.37%, and 15.67% of the genotypes, respectively. Cluster I, had a maximum representation of *indica* type along with six tropical japonica genotypes, namely, IRGC51498, IRGC53087, IRGC5390, IRGC54203, IRGC53089, and IRGC54201. These tropical *japonica* genotypes had more genetic similarities with the short grain aromatic varieties that originated from Kashmir, namely Rahmann Batti P, KEW, and Kal Brar. However, other tropical japonica genotypes were distributed in clusters II, III, and IV. Genotypes originating from the United States, Indonesia, and Philippines were predominantly from clusters II and III, whereas cluster IV was dominated by genotypes from the United States, Liberia, and Taiwan. A graphical representation of genetic distance among the germplasm panel based on neighbour-joining phylogenetic tree (NJ-tree) is given Appendix A.

### 3.4. Population Structure

Population stratification of the total genotype assembly indicated the existence of a distinct structure. For the assessment, we used an assumed population range of K = 2 to K = 8, with three replications per K. The whole population could be divided into two major subpopulations, based on the ΔK value of 603.3 (Table 5). The analysis also indicated a second peak at ΔK of 83.3, suggesting a substructure among the major groups. The grouping of genotypes to any sub-population was performed when their inferred ancestry coefficient exceeded 0.5. Furthermore, genotypes were treated as admixtures where the affiliation probabilities were <90%, whereas genotypes with inferred ancestry ≥90% were classified as unmixed. In the entire assembly, the first subpopulation contained 189 genotypes accounting for 80.0% of the total. This group contained 188 tropical *japonica* accessions plus one genotype, Rahmann Batti P from Kashmir. Moreover, this population contained 139 unmixed genotypes, while the remaining 50 were admixed. Rahmann Batti P was found to be included among the admixed set. The second sub-population possessed 47 genotypes (20% membership proportion) that included most of the *indica* lines, *aus*, and wild rices. Twelve tropical *japonica* lines were also found to be included in the second sub-population, among which ten were admixed. Thirty-four unmixed lines were found in this sub-population, which included two tropical *japonica* accessions, IRGC54201 and IRGC53089. The admixed genotypes of the second sub-population contained non-tropical *japonica* genotypes, such as Taraori Basmati, Kal Brar, and Kew. A further analysis of the substructure among the tropical *japonica* lines revealed the existence of three sub-populations within. This first sub-population (POP1) included 103 lines that could be divided into 83 unmixed lines and 20 admixed lines. The second sub-population (POP2) contained 32 unmixed lines and 13 admixed lines, while the third included 30 unmixed and 22 admixed lines. Integrating the sub-population structure of the whole assembly and that of the tropical *japonica* lines, we could finally divide the entire assembly into four sub-populations: POP1, POP2, and POP3, which included tropical *japonica*s, and POP4, containing *indica* and other lines. Examining each of these sub-populations, we could see that POP1 accounted for 42.8% of the assembly accommodating 101 lines, of which 86 were unmixed and 15 were admixed. The second sub-population, POP2, accounted for 19.1% of the assembly, with 45 genotypes, which were further found to contain 32 unmixed lines and 13 admixed lines. The sub-population, POP3, had a strength of 46 genotypes, among which 27 were unmixed and 19 were admixed types. The last group forming POP4 had 40 constituent genotypes, of which 30 were unmixed and 10 were admixed (Figure 4). Two genotypes did not fall into any sub-populations, viz. Rahmann Batti P and IRGC30921. While the former had major allelic contributions from POP1 and POP4, the latter was a major admixture of POP1 and POP2.

### 3.5. Population Distribution and Differentiation Based on Geographic Origin

A lookup for the distribution of genotypes sourced from different countries revealed that POP1 had an exclusive combination of 16 countries accounting for 35.6% of the total source nations (Figure 5). Six genotypes were exclusive to the POP2, while POP3 and POP4 had four and one exclusive genotypes, respectively. Most of the countries that exclusively contributed to the POP1 genotypes were from the Americas, Europe, Africa, and the Pacific region. POP1 and POP2 shared genotypes from South Korea and Sri Lanka, and genotypes from five countries from the Americas (Bolivia, Colombia, Ecuador, Suriname, and Venezuela) were present in POP3. Additionally, the genotypes sourced from Bhutan, Brazil, Madagascar, and the Philippines were shared among POP1, POP2, and POP3. POP1 also shared genotypes from China with POP2 and POP4. Similarly, genotypes from Japan were found to be grouped among POP1, POP3, and POP4. Four countries, Indonesia, Liberia, Thailand, and the United States, contributed genotypes that were grouped in all four subpopulations. Other than the above combinations, POP2 and POP3 shared genotypes from Ghana. POP4 shared the least similarities from countries that contributed to the remaining subpopulations.

The comparison of the pair-wise fixation index (F_ST_) based on the geographical origin of the population revealed a significant differentiation pattern. The heatmap of pairwise F_ST_ values among the 41 countries of tropical japonica genotypes studied is shown in Figure 6. Among these lines, the pairwise F_ST_ values showed significant differentiation among all of the pairs of sub-population, ranging from 0.00 to 0.91 with an average value of 0.31, suggesting the uniqueness of each group.

Relatively low population differentiation could be seen among the genotypes from Asian countries, viz. Bhutan, China, Thailand, Indonesia, Philippines, Sri Lanka, and Malaysia. Additionally, found among these sets of populations are the genotypes from Brazil, United States, Ghana, Liberia, Guinea, and Madagascar. Among these populations, F_ST_ values ranged below 0.25. Additionally, most of the genotypes of Asian origin showed relatively less differentiation with the genotypes derived from Africa, Europe, the Pacific region, and some parts of the Americas. However, the genetic differentiation between the genotypes derived from African regions such as Kenya, Guyana, Sierra Leone, Guinea-Bissau, Zimbabwe, Zaire, and Gabon differentiated among themselves as well as from those derived from Europe (Italy, Turkey, and France), the Americas (El Salvador, Cuba, Haiti, and Costa Rica), and the Pacific region (Australia, Ponape Island, and Papua New Guinea). Interestingly, the genotypes from Bangladesh were also found to be highly differentiated. An AMOVA was carried out to explore the relative contribution of diverse factors affecting the observed genetic variability and to assess the variation among and between the populations, with each factor considered in different analyses, i.e., species (*indica* and *japonica*) and location (source or different countries). In the total location-wise genetic variance, 18.92% was explained by populations based on structure, and the remaining 81.08% was attributed to variation within populations (Table 6). In contrast, for the genetic variance among *indica*–*japonica* genotypes, 26.78% of the variation was found among the population and 73.22% appeared within the population.

### 3.6. Chromosome-Wise Allele Distribution among the Subpopulations

Among the different chromosomes, a remarkable difference in allele distribution between subpopulations was found (Figure 7). POP4 was found to accumulate more unique alleles that were not found among other subpopulations. There were 14 unique alleles distributed across 6 chromosomes. POP3 had four unique alleles, while POP1 and POP2 has one unique allele each. Analysing the shared alleles, POP4 had more common alleles shared between POP3 than any other subpopulations. Furthermore, chromosomes 5, 8, 10, and 11 had the maximum differentiating alleles for POP4 compared with the rest. Among countries, the genotypes sourced from the United States had the maximum number of alleles that are found common in all four sub-populations. This was followed by the genotypes from Indonesia, Japan, the Philippines, and India.

## 4. Discussion

The information on the gene diversity in plant genetic resources provides an important basis for crop improvement [25], essential to sustaining a high level of productivity [26], and offers opportunities for breeders to produce superior varieties/hybrids with the best combination of economic traits. Indian rice germplasm possesses rich diversity among the *indica* subgroup, which has provided a breakthrough in production and productivity that is confined to the subspecies. Similarly, limitations are also observed in the other major sub-group, *japonica*. It has been demonstrated that assembling the genetic diversity of both sub-species through breeding could result in harnessing unexploited heterosis, which is naturally hindered due to the evolutionary cross incompatibility between *indica* and *japonica* [27]. Although rice hybrids are a reality today, the heterosis levels realized are not convincing enough for outright adoption by farmers. One of the major reasons for low heterosis is the poor heterotic diversity among the parental lines within each of the subspecies.

Hybridization between *indica* and *japonica* has been the subject of rice research for a long time. Several breakthroughs have been reported, such as the identification of widely compatible genes, the development of new plant type (NPT)-based breeding lines [28], and the use of tropical *japonica* as the bridge between them [29]. NPT lines characterized by low tillering capacity, possessing large panicles inherited from the tropical *japonica* gene pool, and carrying widely compatible genes are capable of hybridization with *indica* lines are being used for the diversification of the parental genetic base in *indica* types. Additionally, this opens avenues for the exploitation of heterosis through inter-varietal and interspecific hybrids [30]. Moreover, the utilization of NPT lines provides an opportunity to improve the poor grain filling and the low biomass of interspecific hybrid rice. Notably, several improved rice varieties were developed at the IRRI using NPT lines derived from tropical *japonica* germplasm [31]. Therefore, tropical *japonica* holds the key to improving rice lines, particularly those belonging to the *indica* subspecies, via the development of NPT-based hybrids using information about genetic diversity. Furthermore, the diversification of the parental base is essentially needed in hybrid rice development, as most of the existing cultivar diversity has been tested for combining ability. Combining ability coupled with genetic diversity could lead to the development of heterotic pools in rice that potentially will lay a foundation for the next generation of hybrid rice breeding. Genomic tools, particularly using molecular markers, have become one of the most robust tools used to evaluate genetic variability, genetic structure, and phylogenetic similarities in rice germplasm [32].

In this study, we comprehensively characterized a global collection of tropical *japonica* genotypes for genetic diversity using phenotypic and genotypic features. The geographic locations included 41 countries/territories spread across six continents. Together, 36 rice lines from India, including six wild rice and two *aus* cultivars, were also included in the evaluation. Agronomically, the tropical *japonica* genotypes were highly variable for traits such as tiller number, panicle length, and yield, three major characters used in NPT development. The apparent morphological variation present in grain yield indicates that they harbour variable gene combinations that can give differential yield expressions [33]. Most importantly, the largest contributor to the total variation among the tropical *japonica* lines was panicle length. This implied that NPT development with panicle length as a major yield contributor could be realized from the current population. This was followed by another important yield component: total grain number. A preponderance of high variability for these two traits further consolidates the usability of the tropical *japonica* lines directly into introgression breeding with *indica* lines. Third, plant height showed a significant contribution to total variation, indicating the opportunity to select a suitable plant height while breeding for genotypes adaptable to various geographical locations. For instance, taller genotypes may be preferred if adaptation is sought for problem areas, such as those with water stagnation as a characteristic feature, while semi-dwarf types are suitable for irrigated ecosystems [34]. One of the major traits that are preferred in NPT development is tiller number. Optimal tillering, not too low and not too many, is desirable when ensuring adequate partitioning of metabolites into grains during the grain-formation stage. Excess tillers or late tillers may divert the key metabolites to growth instead of grain formation, which may render panicles under development or may reduce the number of spikelets [35]. Therefore, NPT is an optimized combination of tiller number, panicle length, and grain number leading to high grain yield [36]. Other traits such as spikelet fertility could also play a significant role in yield; hence, high spikelet fertility among NPT lines is the most desirable trait for recruitment into breeding. We could realize all of these features in the current panel, both through variability investigations on individual traits as well as through multivariate analyses using PCA. When grouped, the tropical *japonica* lines were stratified into six clusters based on the total phenotypic variability, wherein 75.8% of the total variation was explained by PC1, PC2, and PC3. Furthermore, we observed that all of the desirable traits of NPT were available in the panel as major contributors of variability. Messmer et al. [37] mentioned that, if the variation is higher or equal to 25%, clustering can be performed to show the similarities among the genotypes. In our findings, traits such as spikelet fertility, yield, tiller number, and panicle length had adequate variability to contribute to the major principal components that could adequately scatter the genotypes in a two-dimensional plot. 

One of the striking observations in the genotype grouping based on yield and the related trait is the incongruity of origin and the clusters. It was seen that all of the clusters had genotypes that originated from countries irrespective of the continents. Although this random pattern could be coincidental, scientific evidence strongly supported the movement of rice from its centre of origin to different parts of the world [38]. Evidence confirmed that the rice originated from China and spread throughout the world through the Indian route, which has further transformed itself as the secondary centre of origin, particularly accumulating *indica* subspecies [39]. Therefore, it is prudent to conclude that tropical *japonica* lines sourced from different parts of the world in this study have moved from southeast Asia. We have seen more evidence for this hypothesis from marker-based genetic diversity. Since the inception of IRRI in the 1960s, a lot of rice germplasm has transferred across continents, accelerating the genetic migration of rice. The Philippines, as the host nation of IRRI, has become pivotal to the rice genetic network across the world. Some good phenotypic characters observed among the tropical *japonica* lines sourced from the Philippines and the United States underpins the role of genetic networks in the modern era because the United States neither uses rice as its staple cereal nor has natural genetic diversity. 

The development of heterotic pools for hybrid rice breeding programmes requires precise evaluation of genome-wide genetic diversity. Earlier, IRRI published a set of the 50 most informative rice microsatellite markers that are robust enough to divulge rice genetic diversity known as GCP markers after the Generation Challenge Programme of Consultative Group for International Agricultural Research (CGIAR) [15]. Of the 50 markers, 46 were polymorphic (92%) in the germplasm assembly and produced as many as four alleles per locus, which indicated a high resolving power of the markers for genetic diversity, even within a section of tropical *japonica* lines. A similar average number of alleles/markers was reported earlier by Anandan et al. [40], who examined 629 rice genotypes, including tropical *japonica* and *indica* accessions using 39 genetic markers. However, an average number of alleles of 2.42 alleles/locus was reported from a set of 100 iso-cytoplasmic restorer lines derived from the 25 commercial hybrids using GCP markers (reference). This further suggested that a significant portion of genetic variability within the germplasm assembly used in this study could be exposed by the GCP markers, indicating their suitability as the smallest and quickest set of explorative markers for genetic diversity investigations in rice. According to DeWoody et al. [41], if the PIC value is more than 0.5, the marker will be considered highly informative, such as six markers identified in this panel, while the average PIC remained as high as 0.36. Several researchers observed similar results using SSR markers in rice [42,43,44], whereas much higher PIC values were also reported in some other studies [45,46,47].

Different grouping patterns have emerged from phenotypic and genotypic data, with altered orientations and distributions of genotypes. This pattern is not unexpected because the genotypic data were derived from random SSR markers and the phenotypic data were based on a few yield and related traits. Unless there is a close association between marker and trait, it is unlikely that both data types can produce a cognizable resemblance. Moreover, the quantitative traits under the influence of the changing environment can also introduce ambiguity in the distribution pattern within and between clusters, whereas such an uncertainty does not exist with the data obtained from genetic markers as they are free from environmental influence and maintains high accuracy and repeatability in the prediction pattern. The genetic architecture of modern-day rice genotypes is designed by processes related to domestication, geography, and breeding.

New plant type (NPT) was a concept developed in rice breeding targeted at improving rice yield by 20–25%. The NPT integrates phenotypic traits such as short, thick, and sturdy stems; dark green leaves; few tillers with long panicles; and high grain numbers. A subsequent survey in the rice germplasm for donors of these traits landed on several tropical *japonica* (*bulu*) lines from Indonesia. Although the first-generation crosses with the NPT donors resulted in high sterility, the subsequent development of second-generation hybrids could improve fertility to a great extent. Ever since Oka [48] described the affinity of genotypes belonging to *indica* and *japonica* towards articulating hybrid fertility, *indica**–japonica* hybridization has been a major breeding interest in rice. Although earlier attempts to make hybrids had met limited success, the NPT lines in the tropical *japonica* background renewed the interest in the development of new hybrids [49]. In addition, the discovery of a wide compatibility (WC) system [50] and the consequential identification of WC genes have contributed ways to improving fertility in hybrids [51], particularly in super rice hybrids in China. Therefore, tropical *japonica* germplasm provides greater promise in parental line development, diversification, and hybrid breeding in rice. One of the major requirements for hybrid development is the documentation of genetic variability and population structure. The advent of molecular systems in the late 1980s has triggered research on deciphering the diversity pattern in rice. Glaszmann [52] used isozyme variability to pattern the genetic relationship among a large collection of 1688 traditional Asian rice lines and divided them into six varietal groups, two major, two minor, and two satellite groups. Subsequently, several studies on genetic diversity and population structure in rice have been published using DNA-based molecular markers [53,54,55,56,57] that used germplasm from different regions. Garris et al. [12] grouped a large number of accessions into five major clusters such as *aus*, temperate *japonica*, *indica*, aromatic, and tropical *japonica*. Likewise, this study assessed diversity at the molecular level among tropical *japonica* lines collected worldwide, together with a few *indica*, *aus*, and wild accessions from India. We could classify the germplasm panel used into four distinct subpopulations, which included two major groups, separating *indica* and tropical *japonica* initially, with further sub-structuring of the tropical *japonica* set into three subgroups. Taken together, sub-populations derived from tropical *japonica* are designated as POP1, POP2, and POP3, while the population that included *indica* accessions is POP4. Furthermore, the population structure identified in this study showed concordance with the distance-based clustering from the principal coordinate analysis (PCoA). PCoA is an method extensively utilized to evaluate genetic diversity based on quantitative and qualitative traits, which scales the distance data into the multidimensional planes to characterize diversity. However, the grouping based on population structure seems to be more accurate, as it could precisely differentiate the *indica* and *japonica* types. The only exception was Rahmann Bhatti P, a landrace from Kashmir, which seems to be a perfect admixture between *indica* and *japonica*. Another notable feature was the placement of wild rice among POP4, indicating that cross-breeding occurred in the wild accessions naturalized under Indian conditions. Citing similar reasons, Sun et al. [58] have reported that most wild rice populations in South Asia have closer similarity to the *indica* subspecies. 

However, we could make a most interesting observation in the distribution of genotypes based on the country of sourcing and their distribution among the subpopulations. Pair-wise F_ST_ values showed a lower level of differentiation between genotypes from Asia and those from the rest of the world, indicating that all of the genotypes sourced outside the centre of origin could be migrants that moved out at different times. Nevertheless, significant differentiation also could be noticed among the tropical *japonica* accessions sourced from some countries of Africa, America, Europe, and Pacific regions, suggesting independent evolution of population within the geographical confines after migrating from the centre of origin. Moreover, the distribution of source countries among the subpopulations indicated no particular pattern within the tropical *japonica*. POP1 being the largest subpopulation accumulated genotypes drawn exclusively from 16 countries, further suggesting a common origin of the genotypes. Moreover, the inclusive distribution of source countries between subpopulations was common within tropical *japonica*, while the *indica* types sourced from India were confined to only one subpopulation, POP4. This pattern could be due to the major representation of tropical *japonica* genotypes in the germplasm assembly, wherein the *indica* types occupied merely 16% of the population. 

To conclude, the current study revealed the population structure of a set of tropical *japonica* accessions sourced worldwide, which indicated dispersion from a common region in Asia. Most of these tropical *japonica* lines possessed NPT characteristics in various combinations, indicating their potential usefulness in pre-breeding to develop superior parental lines for hybrid development as well as in recombination breeding with *indica* types. Particularly, traits such as long panicles and high grain numbers found among several lines could be useful for breeding for higher yield. With a high level of crossability of some of these lines with an *indica* genotype, Pusa 44 has already been assessed [7,9] and the presence of fertility-restoring genes *Rf3* and *Rf4* for the wild-abortive (WA) cytoplasm has been reported. It would be of interest to assess the level of WC properties in these lines. The consolidation of genetic characteristics hitherto available on these tropical *japonica* lines sheds light on their potential use in rice improvement, especially in parental line development for hybrid breeding as well as for recombination breeding.

## Figures and Tables

**Figure 1 genes-13-00484-f001:**
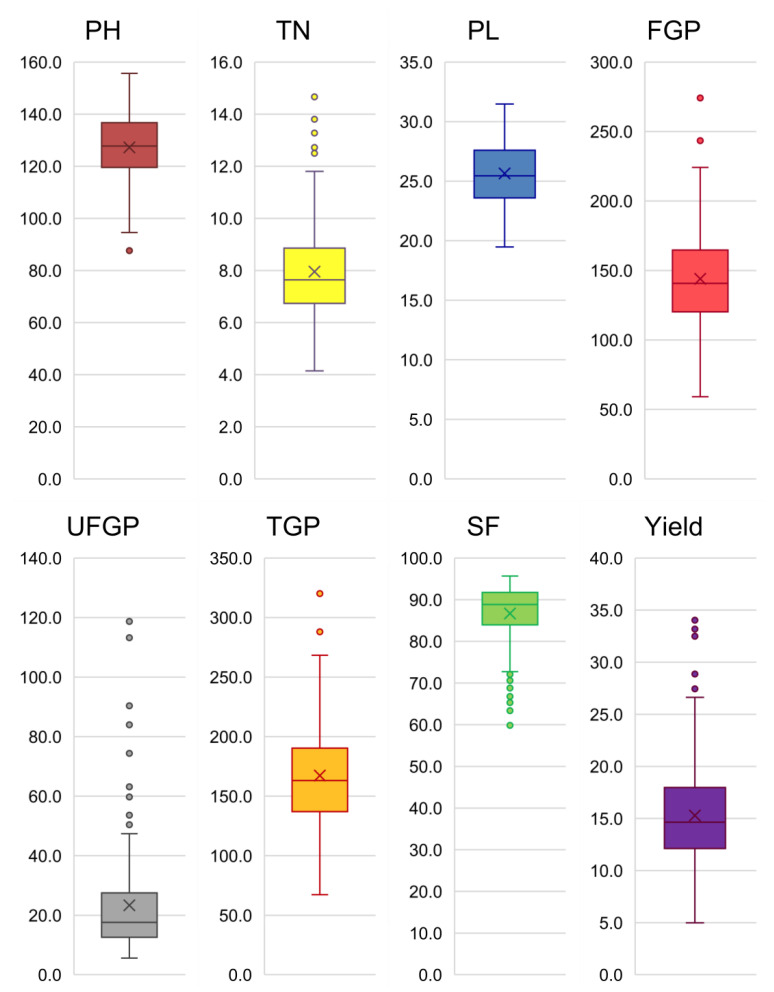
Box plots of agro-morphologic traits among the tropical *japonica* lines. Originating from 43 countries, the genetic variability among the lines was relatively low.

**Figure 2 genes-13-00484-f002:**
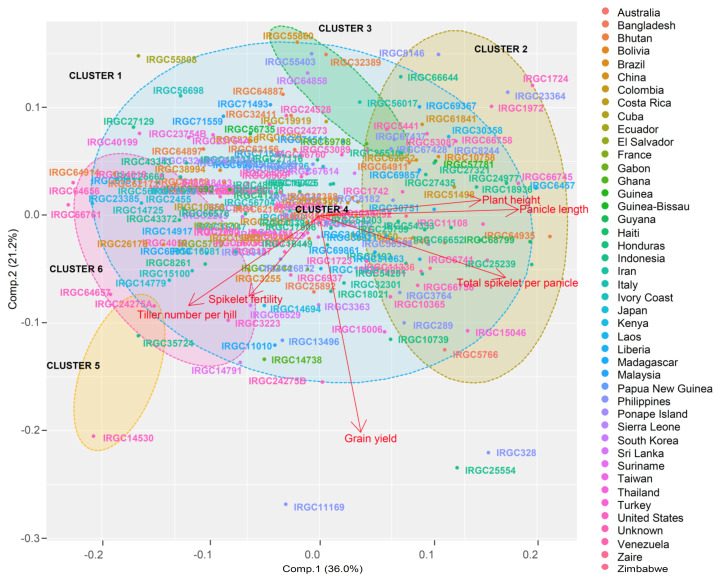
PCA based on agronomic data. Panicle length, grain number, plant height, and spikelet fertility accounted for the maximum variability along with the first principal component, while grain yield and tiller number accounted for the variation along the second principal component axis. The grouping of the genotypes showed at least five clusters with maximum concentration in one cluster. The geographical distribution across clusters showed no definite pattern, indicating that the tropical *japonica* genotypes had a common origin and were distributed across the globe through domestication activities.

**Figure 3 genes-13-00484-f003:**
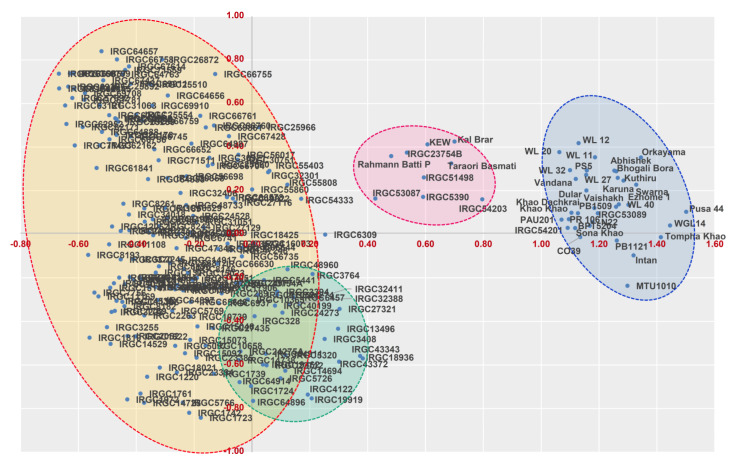
Principal coordinate analysis of 200 tropical *japonica* lines and 36 check lines based on 46 SSR markers.

**Figure 4 genes-13-00484-f004:**
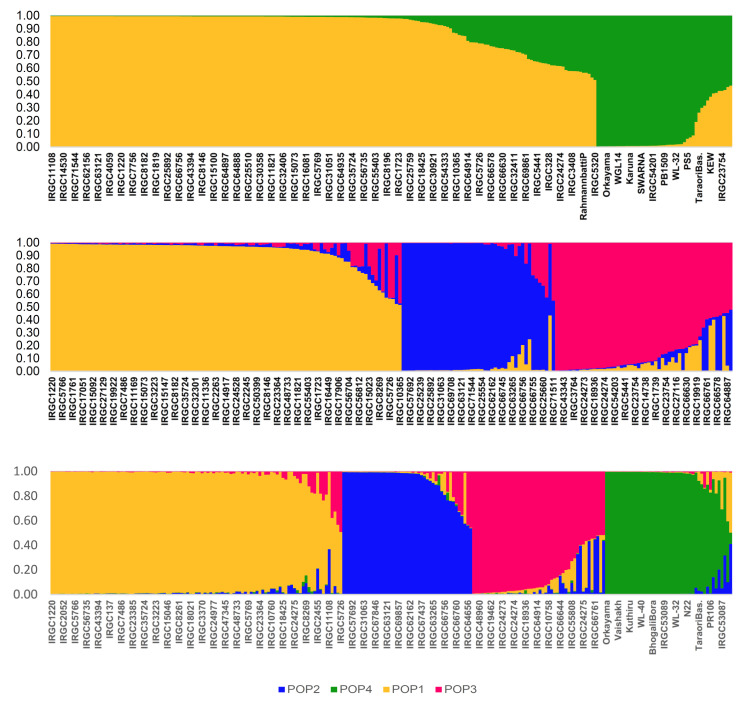
Estimated population structure of 236 rice accessions, which included 200 tropical *japonica* lines.

**Figure 5 genes-13-00484-f005:**
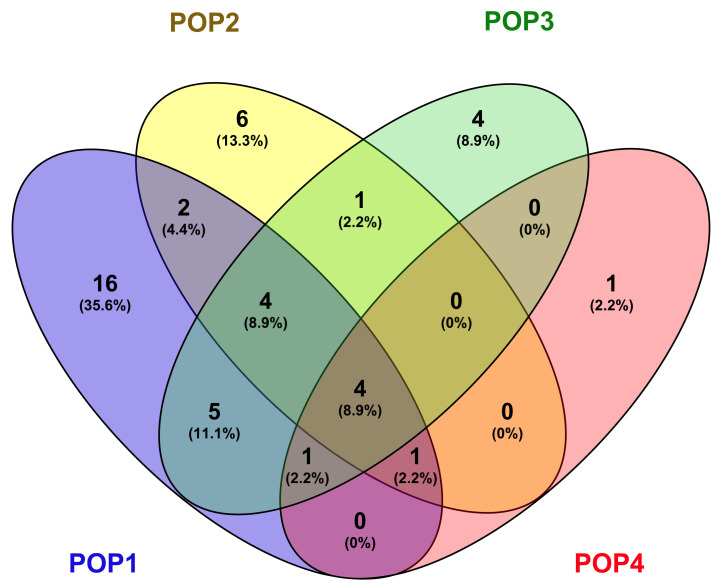
Distribution of sources of origin of germplasm lines among the four sub-populations.

**Figure 6 genes-13-00484-f006:**
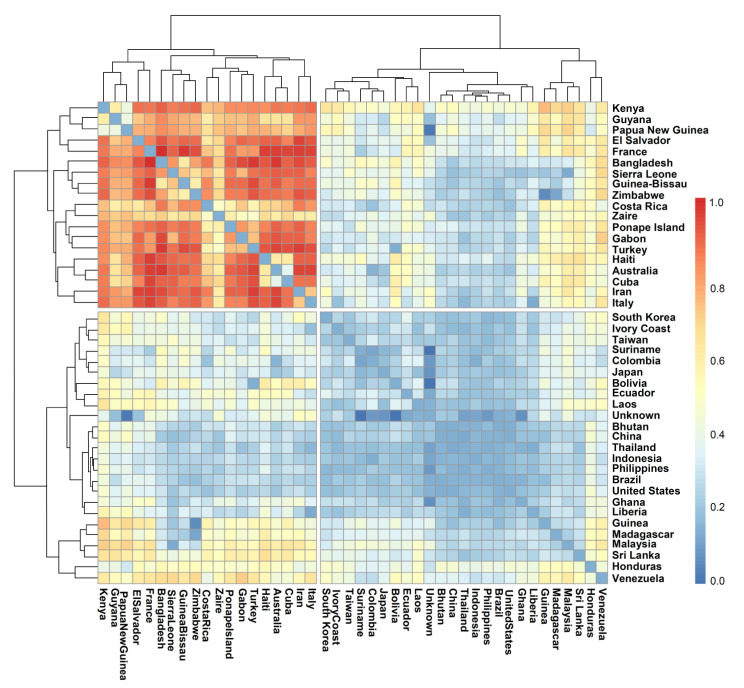
Pairwise FST matrix of tropical *japonica* lines based on their countries of origin. The population showed greater differentiation among 19 countries, while the rest of the population showed relatively little genetic difference.

**Figure 7 genes-13-00484-f007:**
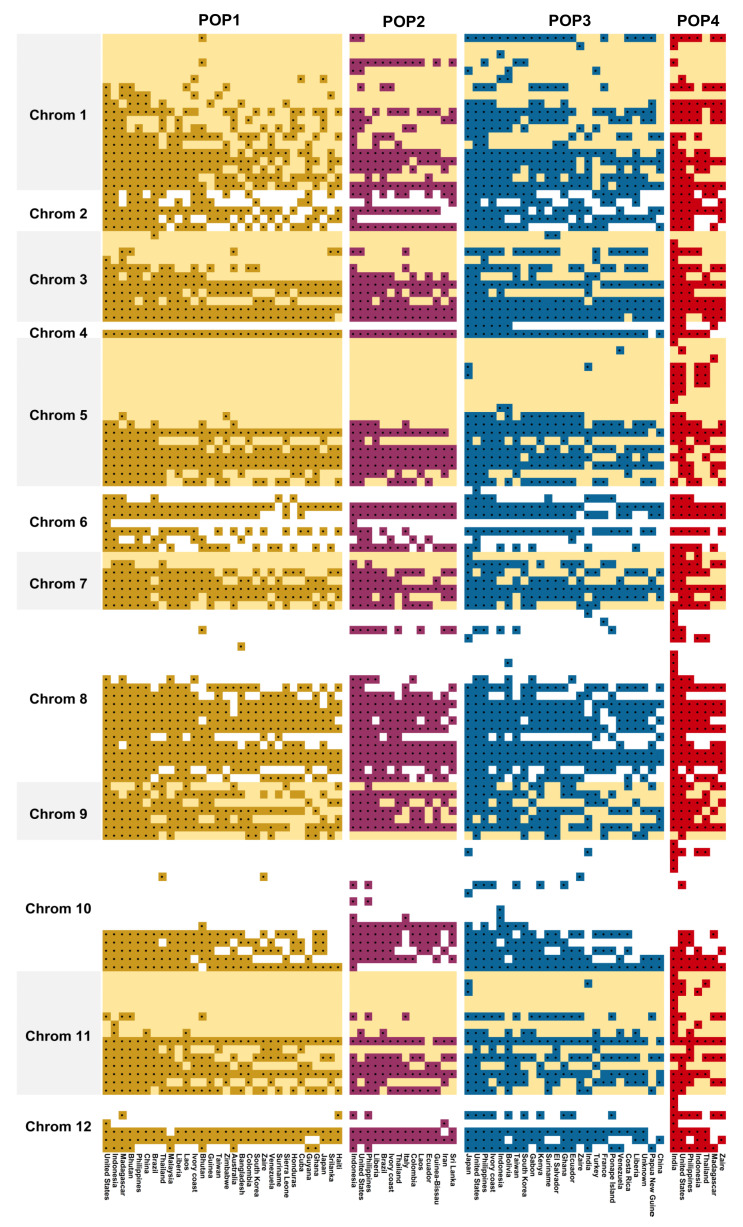
Chromosome-wise allele distribution among the genotypes of four subpopulations sourced from different countries.

**Table 1 genes-13-00484-t001:** Analysis of variance for the morphological traits among the tropical japonica lines grown during three consecutive seasons (environments) at New Delhi.

Source	Df	Mean Sum of Squares							
		PH	TLN	PL	FG	UFG	TG	SF	YLD
Genotype (G)	199	1000.2 **	21.1 **	43.9 **	6929.0 **	1784.7 **	10007.0 **	305.2 **	142.4 **
Year (Y)	2	106.0	1.3	0.8	481.0	150.2	111.0	136.1	33.5
G × Y	398	481.5 **	12.7 **	14.5 **	2946.0 **	569.7 **	3900.0 **	131.1 **	42.1 **
Rep (Year)	3	2410.3	34.3	219.8	11745.0	932.9	19070.0	62.1	311.8
Residual	597	144.4	2.15	20.8	160.0	34.4	201.0	9.5	3.3
CV (%)		15.8	37.5	18.7	32.8	96.2	33.3	11.5	41.4
LSD (0.05)		2.2	0.3	0.8	2.3	3.4	2.6	0.6	0.3
Mean		127.3	8.0	25.6	144.0	23.4	167.4	86.7	15.3
Minimum		87.7	4.2	19.5	59.1	5.6	67.2	59.8	5.0
Maximum		155.7	14.8	31.5	274.1	118.7	323.8	95.7	34.1

** *p* ≤ 0.01; Df, degrees of freedom; PH, plant height in cm; TLN, number of tillers; PL, panicle length in cm; FG, number of filled grains per panicle; UFG, number of unfilled grains per panicle; TG, total number of grains per panicle; SF, spikelet fertility in percentage; YLD, grain yield per plant in g.

**Table 2 genes-13-00484-t002:** Statistics of the principal components obtained from the agro-morphological traits among the tropical *japonica* lines. The contribution of different traits to major principal components is given in boldface.

Statistics		PC1	PC2	PC3	PC4	PC5	PC6
Standard deviation		1.47	1.12	1.06	0.89	0.66	0.46
Proportion of Variance		0.36	0.21	0.19	0.13	0.07	0.04
Cumulative Proportion (%)		36.0	57.2	75.8	89.1	96.5	100.0
Eigenvalues		2.16	1.27	1.12	0.79	0.44	0.21
Trait contribution (%)	PH	21.17	0.32	0.04	54.39	23.86	0.21
	TLN	13.80	12.08	35.36	11.99	0.35	26.43
	PL	31.79	0.05	0.43	3.28	62.65	1.81
	TG	27.92	5.99	0.35	25.69	12.12	27.93
	SF	3.92	9.91	63.69	4.61	0.94	16.95
	YLD	1.41	71.66	0.13	0.05	0.08	26.67

PC, principal component; PH, plant height in cm; TLN, number of tillers; PL, panicle length in cm; TG, total number of grains per panicle; SF, spikelet fertility in percentage; YLD, grain yield per plant in g.

**Table 3 genes-13-00484-t003:** Centroids of agro-morphological data of tropical *japonica* lines based on phenotypic divergence with standard error.

Cluster	Membership %	Centroid ± Standard Error					
		PH	TLN	PL	TG	SF	YLD
1	86.5	127.9 ± 1.0	7.8 ± 0.1	25.9 ± 0.2	165.0 ± 2.6	88.6 ± 0.35	15.5 ± 0.4
2	7.5	137.3 ± 2.7	7.8 ± 0.5	28.1 ± 0.5	246.3 ± 10.7	73.4 ± 2.34	16.1 ± 1.1
3	2.5	142.4 ± 3.6	8.5 ± 0.4	26.0 ± 1.0	121.2 ± 12.0	70.9 ± 2.20	8.2 ± 0.6
4	0.5	102.0 ± 0.0	10.8 ± 0.0	24.8 ± 0.0	238.8 ± 0.0	58.9 ± 0.0	20.1 ± 0.0
5	1.5	95.3 ± 5.2	11.4 ± 0.2	23.8 ± 1.1	136.7 ± 7.8	87.7 ± 1.18	25.0 ± 3.9
6	1.5	131.9 ± 8.3	15.5 ± 0.9	24.7 ± 1.0	109.1 ± 23.1	91.8 ± 1.31	14.6 ± 2.9

PH, plant height in cm; TLN, number of tillers; PL, panicle length in cm; TG, total number of grains per panicle; SF, spikelet fertility in percentage; YLD, grain yield per plant in g.

**Table 4 genes-13-00484-t004:** Marker statistics based on 46 GCP panel microsatellite markers among the tropical *japonica* lines.

Chrom	Loci	Allele Frequency			AF_m_	H_e_	PIC	A_e_	Rare Alleles
		Total	Average	Class					
1	6	19	3.17	2, 3, 6	0.689	0.423	0.366	1.863	1
2	2	5	2.50	2, 3	0.603	0.519	0.432	2.114	-
3	5	11	2.20	2, 3	0.761	0.294	0.237	1.528	-
4	1	2	2.00	2	0.931	0.128	0.120	1.147	-
5	5	18	3.60	2, 3, 4, 6	0.705	0.405	0.353	1.848	2
6	3	8	2.67	2, 4	0.772	0.298	0.250	1.579	1
7	3	7	2.33	2, 3	0.615	0.455	0.380	2.002	-
8	7	21	3.00	2, 3, 4	0.669	0.400	0.338	1.888	2
9	3	7	2.33	2, 3	0.693	0.432	0.360	1.786	-
10	3	16	5.33	2, 3, 11	0.566	0.511	0.436	2.261	5
11	4	15	3.75	3, 4, 5	0.714	0.402	0.345	1.734	-
12	2	7	3.50	3, 5	0.772	0.361	0.312	1.565	2

Chrom, chromosome; AF_m_, major allele frequency; H_e_, expected heterozygosity, PIC, polymorphic information content; A_e_ = effective number of alleles.

**Table 5 genes-13-00484-t005:** Simulation statistics for determining the optimal structure of the tropical *japonica* population.

K	Reps	Mean LnP(K)	SD LnP(K)	Ln’(K)	|Ln”(K)|	Delta K
1	3	−12876.9	0.2	-	-	-
2	3	−10938.2	2.2	1938.7	1334.5	603.3
3	3	−10334.1	27.0	604.2	35.8	1.3
4	3	−9765.7	3.6	568.3	301.7	83.3
5	3	−9499.1	13.7	266.6	46.6	3.4
6	3	−9279.1	62.9	220.0	26.9	0.4
7	3	−9086.0	31.6	193.1	91.2	2.9
8	3	−8984.0	63.5	102.0	-	-

K, number of assumed subpopulations; reps, replications; SD, standard deviation.

**Table 6 genes-13-00484-t006:** Analysis of molecular variance (AMOVA) of 200 tropical japonica lines including checks based on three parameters, origin, subspecies, and subpopulations.

Particulars	Source of Variation	Df	SS	Variance	Percentage of Variation	
Country of origin	Among populations	25	1164.52	46.58	1.93	0.19
	Within populations	210	2817.14	13.41	5.26	0.52
	Individuals within Populations	236	684.00	2.90	2.90	0.29
	Total	471	4665.67		10.09	1.00
	Fixation Index (F_ST_)	0.19				
Sub-species	Among populations	3	493.07	164.36	3.55	0.28
	Within populations	232	3491.13	15.05	6.12	0.49
	Individuals within Populations	236	664.00	2.81	2.81	0.23
	Total	471	4648.20		12.48	1.00
	Fixation Index (F_ST_)	0.28				
Populations	Among populations	3	478.25	159.42	1.30	0.13
	Within populations	232	3505.95	15.11	6.15	0.60
	Individuals within Populations	236	664.00	2.81	2.81	0.27
	Total	471	4648.20		10.26	1.00
	Fixation Index (F_ST_)	0.127				

Df, degrees of freedom; SS, sum of squares.

## Data Availability

The data presented in this study are available in the Appendix A.

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
