# Peer review of "Population Structure of a Worldwide Collection of Tropical Japonica Rice Indicates Limited Geographic Differentiation and Shows Promising Genetic Variability Associated with New Plant Type"

_genes, 2022, doi:10.3390/genes13030484_

Round 1
Reviewer 1 Report
The manuscript entitled " Population Structure of a Worldwide Collection of Tropical Japonica Rice Indicates Limited Geographic Differentiation and
Shows Promising Genetic Variability Associated with New Plant Type" characterized a set of tropical japonica rice lines to identify phenotypic and genotypic diversity and population structure by using SSR markers and agro-morphological traits. The study is very meaningful for identifying potential parental sources for future hybrid rice breeding. The data were well analyzed and displayed. I suggest to accept the manuscript just after minor revising on English language.
Author Response
Response 1.
We thank the reviewer for his/her encouraging comment. We have rechecked the entire manuscript for readability, language and accuracy of data presents. We have made appropriate corrections as required. We are confident that the manuscript has improved significantly based on the reviewers suggestions.
Reviewer 2 Report
Generally, the research was well performed and written in a clear presentation. However, there are some minor things the authors need to perfect before accepted:
- Figure 2 and 3 there are too many dots, the authors need to pick some dots and make it clearer.
- Table 5 the authors mentioned simulation, however, there is no simulation procedure described.
Author Response
Comment 1. Figure 2 and 3 there are too many dots, the authors need to pick some dots and make it clearer.
Response: The figures 2 and 3 are modified by reducing the dot size.
Comment 2. Table 5 the authors mentioned simulation, however, there is no simulation procedure described.
Response: The Bayesian process in the population structure analysis simulates the population statistics based on the k range selected while running the MCMC simulations. The table 5 provided the abstract of the simulation summary.